# Foliar Selenate and Zinc Oxide Separately Applied to Two Pea Varieties: Effects on Growth Parameters and Accumulation of Minerals and Macronutrients in Seeds under Field Conditions

**DOI:** 10.3390/foods12061286

**Published:** 2023-03-17

**Authors:** Maksymilian Malka, Gijs Du Laing, Alžbeta Hegedűsová, Torsten Bohn

**Affiliations:** 1Laboratory of Analytical Chemistry and Applied Ecochemistry, Department of Green Chemistry and Technology, Faculty of Bioscience Engineering, Ghent University, Coupure Links 653, 9000 Ghent, Belgium; 2Institute of Horticulture, Faculty of Horticulture and Landscape Engineering, Slovak University of Agriculture in Nitra, Tr. A. Hlinku 2, 94976 Nitra, Slovakia; alzbeta.hegedusova@uniag.sk; 3Nutrition and Health Research Group, Department of Precision Health, Luxembourg Institute of Health, 1 A-B, Rue Thomas Edison, 1445 Strassen, Luxembourg

**Keywords:** crop biofortification, selenium and zinc, trace element deficiency, nutrition, food supply

## Abstract

Though selenium (Se) and zinc (Zn) constitute essential nutrients for human health, their deficiencies affect up to 15% and 17% of the global population, respectively. Agronomic biofortification of staple crops with Se/Zn may alleviate these challenges. Pea (*Pisum sativum* L.) is a nutritious legume crop that has great potential for Se/Zn biofortification. Herein, two varieties of pea (Ambassador, Premium) were biofortified via foliar application of sodium selenate (0/50/100 g of Se/ha) or zinc oxide (0/375/750 g of Zn/ha) during the flowering stage under field conditions. While no significant differences were found in Se accumulation between seed varieties upon Se treatments, selenate enhanced the accumulation of Se in the two seed varieties in a dose dependent manner. Selenium concentration was most elevated in seeds of Ambassador exposed to 100 g of Se/ha (3.93 mg/kg DW compared to the control (0.08 mg/kg DW), *p* < 0.001). 375 g of Zn/ha (35.7 mg/kg DW) and 750 g of Zn/ha (35.5 mg/kg DW) significantly and similarly enhanced Zn concentrations compared to the control (31.3 mg/kg DW) in Premium seeds, *p* < 0.001. Zinc oxide also improved accumulations of Fe, Cu, Mn, and Mg in Premium seeds. Se/Zn treatments did not significantly affect growth parameters and accumulations of soluble solids and protein in seeds. Positive and significant (*p* < 0.01) correlations were observed between Zn and Fe, Cu, Mn and Mg levels in Premium seeds, among others. Consuming 33 g/day of pea biofortified with Se at 50 g/ha and 266 g/day of pea biofortified with 375 g of Zn/ha could provide 100% of the RDA (55 μg) for Se and RDA (9.5 mg) for Zn in adults, respectively. These results are relevant for enhancing Se/Zn status in peas by foliar biofortification.

## 1. Introduction

Selenium (Se) and zinc (Zn) are indispensable micro-elements in the human body. Selenium (as selenoproteins) and Zn act as cofactors for many important antioxidant enzymes, such as glutathione-peroxidase (GPx) and superoxide-dismutase (SOD), respectively [1,2]. Deficiency of Se and Zn can lead to increased risk of mortality and development of multi-factorial illnesses, both communicable and non-communicable [1,3,4,5,6]. However, higher intakes have also been reported to produce adverse health effects, as reviewed previously [7,8].

Selenium and Zn deficiencies affect up to 15% and 17% of the global population, respectively [9,10]. It should be stressed that both deficiency and suboptimal status in micronutrients negatively affect human health. Reports have emphasized widespread below-optimal levels of Se and Zn throughout Europe. This is largely attributable to inadequate concentrations in soil [11], reflecting depletions by agricultural practices and rainfall. Dietary Se/Zn availability is associated to a great extent with the soil Se/Zn level and the bioavailability of Se/Zn from main crops [12,13,14], though Se/Zn in soil is not homogenous in its distribution and availability to crops [15,16].

Agronomic biofortification is a promising approach that aims to enhance the nutritional value (mostly micronutrients) of staple crops. This strategy relies on optimized application of fertilizers to the soil and/or crop leaves (foliar fertilization), as reviewed previously [17,18,19]. Foliar fertilization is a very efficient way of Se and Zn plant biofortification [20,21]. In addition, under field conditions, foliar fertilization is less damaging to the environment, preventing accumulation of the fortificant in the soil, leading potentially to contamination [20,22]. The efficacy of foliar application of trace elements depends on a number of factors, namely the physico-chemical specifications of the formulation, the environmental conditions during the time of spraying, or the characteristics of the plant—such as leaf size and metabolism—to which spraying is applied, as summarized previously [23].

Although legumes are staple food crops for billions of people worldwide, their biofortification is still insufficiently used as an approach for alleviating hidden hunger [24,25]. Pea (*Pisum sativum* L.) is a cool season legume harvested all over the world, mainly in temperate regions [26]. The global harvest of dry peas in 2020 was reported as 14.6 million tons, with cultured areas spanning 7.2 million ha (FAOSTAT 2022). Pea is an important human and animal food crop that plays critical roles (as do other pulses) in sustainable agriculture, biodiversity and the environment, as well as in global health and food security [27,28,29]. Peas are an important source of protein and carbohydrates such as starch, but also provide dietary fibre, minerals and vitamins, in addition to phytochemicals, among others [30,31]. Pea protein is a high-biological-value protein that exhibits health benefits encompassing antioxidant, antihypertensive, and prebiotic-like effects on the gut microbiota [32]. Consumption of peas and their components has been linked to aspects of cardiometabolic as well as gastrointestinal health [30,33].

Consequently, in this investigation we studied the response of two pea varieties (Ambassador, Premium) to foliar application of Se and Zn during the flowering stage under field conditions. Growth parameters, accumulation of selected trace elements (including Se and Zn), macro elements, soluble solids and protein in seeds, as well as their respective correlations were evaluated and compared with previous findings, including our pot experiment [34,35,36].

## 2. Materials and Methods

### 2.1. Chemicals

Zinkuran SC was obtained from Arysta LifeScience Slovakia s.r.o. (Nové Zámky, Slovakia). Sodium selenate was purchased from Alfa Aesar (Karlsruhe, Germany). HNO_3_ and H_2_O_2_ (suprapure quality) were ordered from LGC Standards (Molsheim, France) and Merck/VWR (Leuven, Belgium), respectively.

### 2.2. Experimental Design and Sample Preparation

A field experiment was carried out in 2014 at the Slovak University of Agriculture in Nitra/Botanical Garden (Nitra, 48.305 N, 18.096 E). The design consisted of four replicates, two varieties of pea (*Pisum sativum* L.) and five treatments, totalling 40 plots. The area of each plot was 1 m^2^. Pea was sown at a rate of 85 seeds / m^2^ at a seeding depth of 5 cm. To minimize potential Se/Zn cross-contamination, border rows were laid out between each plot. Soil analysis and weather data during the growing season (April–June) in Nitra in 2014 are presented in Table 1. Soil element concentrations were measured using the procedure of Varényiová et al. [37] for total N, and available P, K, Mg, and Ca; the methods of Ducsay et al. [38] and Lindsay and Norvell [39] were used to determine total Se and available Zn, respectively. Seeds of two high-yielding and dark-seeded pea varieties were selected for the investigation. Ambassador (representing late variety/restored hybrid) and Premium (constituting an early variety/open pollinated) were obtained from a farmer from the area of Nitra. Selenium (sodium selenate) and Zn (Zinkuran SC (30% ZnO + 6% chelate)) were administered as part of the following treatments: unaltered control, 50 g (Se1), 100 g (Se2), 375 g (Zn1) and 750 g (Zn2), all per ha. The applied solutions had final concentrations of 0.1 and 0.2 g/L of Se or 0.75 and 1.5 g/L of Zn. Foliar application of Se/Zn was conducted at the flowering stage of the pea plants within dry weather periods using a manual sprayer. Additional fertilizers were not employed. Regular irrigation and phytosanitary measures were applied during the experiment. Incidences of pests/diseases and adverse/toxic impacts of foliar Se/Zn applications on plants were not reported. Seeds were harvested at physiological maturity. The soluble solids concentration was determined in fresh seeds, while the remainder of the seeds was rapidly lyophilized and ground to determine the concentration of trace (Se, Zn, Fe, Cu, Mn, Mo) and macro elements (Ca, Mg, K, Na) and protein.

### 2.3. Growth Measures

The number of seeds/pod, and length, perimeter and width of the pod were determined following harvest. Drying of samples was carried out in a drying oven (105 °C) until no more weight changes were noted prior to seed dry matter measurements.

### 2.4. Measured Soluble Solids and Protein

The level of soluble solids (SSC) was determined with a handheld digital refractometer (DR201-95, A. KRÜSS Optronic, Hamburg, Germany). All samples were measured in duplicates; the average was then expressed as the SSC value (% fresh weight (FW)).

Protein level (in duplicates) was measured following the Dumas technique using the TruSpec CHNS analyser (LECO, Saint Joseph, MI, USA). The values obtained were multiplied by the respective nitrogen conversion factor, 5.4 [40], to derive the amount of protein in the investigated samples.

### 2.5. Concentration of Minerals

Following Kaulmann et al. [41], 250 mg of freeze-dried pea was mineralised in 7 mL HNO_3_ (68%) and 3 mL H_2_O_2_ (30%). The mineralisation was done in PTFE vials in a microwave furnace (Multiwave Pro, Anton Paar, Graz, Austria), by raising the temperature and pressure to 200 °C and 30 bars, respectively. Afterwards, samples were diluted to 25 mL with ultra-pure water. A reference vegetable (Spinach, NCS ZC 73013, LGC Standards, Molsheim, France) was employed during each mineralisation cycle. Samples were analysed by ICP-MS (Elan DRC-e, Perkin Elmer, Waltham, MA, USA).

### 2.6. Statistical Analysis

The normality of data distribution (normality plots) and variance equality (box plots) were tested. When needed, the data was logarithmically transformed to better fit a normal distribution. Afterwards, multivariate models were created, with the number of seeds/pod, pod length, pod perimeter, pod width, seed dry matter, and concentrations of soluble solids, protein, Se, Zn, Fe, Cu, Mn, Mo, Ca, Mg, K, and Na in seeds as the dependent observed variables, and pea variety (2 levels) and level of biofortification (5 levels, 2 for Se, 2 for Zn, and controls) as fixed factors. The different levels of biofortification were nested within the biofortificant. Whenever significant results from Fisher-F tests were obtained, all possible group-wise comparisons were conducted (Bonferroni post-hoc tests). When interactions were significant, models were run again by keeping one of the interacting terms constant. *p*-values < 0.05 (2-sided) were regarded as significant. SPSS (vs. 25.0, IBM, Chicago, IL, USA) was chosen for all statistical investigations.

## 3. Results

### 3.1. General Effects

After constructing the multivariate models, combined analysis of variance indicated that treatment (pooled varieties) had a significant impact on the levels of Se, Zn, Fe, Cu, Mn, Mg, and protein in seeds. Variety (pooled treatments) revealed a significant impact on all variables except for the concentration of Se, K, and soluble solids in seeds. The treatment x variety interaction was significant in part (Table 2).

### 3.2. Growth Parameters

Treatment had no significant impact on the number of seeds/pod, length, perimeter and width of the pod, and seed dry matter vs. controls of both varieties. Ambassador had a significantly higher number of seeds/pod, length, perimeter and width of the pod than Premium for all treatments. Ambassador demonstrated slightly but significantly more elevated seed dry matter compared to Premium for all treatments except for Se2 (Figure 1A–E).

### 3.3. Soluble Solid Levels (SSC) in Seeds

Treatment had no significant influence on SSC vs. controls in the two varieties. Premium showed slightly but significantly elevated SSC vs. Ambassador for the Se2 treatment (Figure 1F).

### 3.4. Protein Levels in Seeds

Biofortification did not significantly impact protein levels vs. the control in Ambassador, while Se1 application significantly reduced protein levels vs. the control in Premium. Ambassador exhibited significantly elevated protein levels compared to Premium for all treatment conditions but not for Zn1 (Figure 1G).

### 3.5. Concentration of Trace Elements in Seeds

Selenium biofortification significantly improved Se levels vs. controls (in both varieties). The highest concentration of Se was encountered in Ambassador exposed to Se2 vs. the control, followed by Premium exposed to Se2 vs. the control. Differences found in Se levels between Ambassador and Premium for Se1 and Se2 were not significant. Ambassador had slightly though significantly higher levels of Se than Premium for the control (Figure 2A).

Zinc treatments did not significantly affect Zn levels compared to the control in Ambassador, while Zn1 and Zn2 significantly elevated Zn levels vs. the control in Premium. No significant differences were found regarding the concentration of Zn between Ambassador and Premium for Zn1 and Zn2. Ambassador displayed significantly raised Zn concentration vs. Premium for the control group (Figure 2B).

The Zn2 treatment significantly improved Fe levels vs. the control in Ambassador, while Zn1 and Zn2 significantly raised Fe levels vs. the control in Premium. Ambassador exhibited significantly elevated levels of Fe relative to Premium for all treatments but not for Zn1 (Figure 2C).

Biofortification had no significant impact on Cu levels vs. the control in Ambassador, while Zn1 and Zn2 significantly raised Cu levels vs. the control in Premium. Ambassador had significantly higher Cu levels than Premium for all biofortification treatments except for Zn1 (Figure 2D).

Treatment failed to significantly impact Mn levels vs. the control in Ambassador, while Zn1 and Zn2 significantly raised Mn levels vs. the control in Premium. No significant differences were found regarding concentrations of Mn between Ambassador and Premium (all treatments, Figure 2E).

Treatment had no significant impact on Mo levels vs. controls in both varieties. Premium displayed significantly greater concentrations of Mo than Ambassador for all treatments (Figure 2F).

### 3.6. Concentration of Macro Elements in Seeds

Treatment did not significantly influence Ca levels compared to controls in both varieties. Premium exhibited significantly elevated concentrations of Ca than Ambassador for all treatments (Figure 2G).

Treatment had no significant impact on Mg levels when compared to the control in Ambassador, while Zn1 and Zn2 significantly increased Mg levels vs. the control in Premium. Ambassador had significantly elevated levels of Mg relative to Ambassador for the control, Se1, and Se2 (Figure 2H).

Biofortification did not significantly affect K levels compared to controls in the two varieties. No significant differences were revealed in terms of the concentrations of K between Ambassador and Premium (all treatments, Figure 2I).

The treatment failed to significantly influence Na concentrations compared to controls in both varieties. Premium exhibited significantly elevated levels of Na vs. Ambassador (all treatments, Figure 2J).

### 3.7. Correlations

For Ambassador, a positive and significant correlation was encountered between the number of seeds/pod and pod length and seed dry matter, between pod length and Mn concentration, between pod circumference and pod width, between Se concentration and pod width, between soluble solids and Zn and Mg concentrations, between Zn and Mo and Ca concentrations, between Mn and Fe, and between Ca and K concentrations, and between Ca and Mg and Na concentrations. A negative and significant correlation was encountered between the number of seeds/pod and pod width, and between Se and Mo concentrations. For Premium, a significant and positive correlation was found between the number of seeds/pod and pod length, pod circumference, pod width and Zn concentration, between pod circumference and pod width, between pod length and protein concentration, between seed dry matter and soluble solid and Se concentrations, between Zn and Fe, Cu, Mn and Mg levels, between Fe and Cu, and between Mg and Mn levels, between Cu and Mn and Mg levels, between Mn and Ca and Mg concentrations, and between K and Na concentrations. A negative and significant correlation was observed between seed dry matter and the number of seeds/pod, pod length and protein concentration, between Ca concentration and pod circumference and pod width, between Na and protein concentrations, between Se and Zn and Cu levels, and between Se concentration and the number of seeds/pod (Figure 3).

## 4. Discussion

In this study, we scrutinized the consequences of the foliar application of selenate and zinc oxide on two *Pisum sativum* L. varieties (Ambassador, Premium) during the flowering stage. Parameters related to growth and concentrations of several trace elements (including Se and Zn), macro elements, and macronutrients were determined in seeds. Selenate improved the concentration of Se in both pea varieties (Table 2, Figure 2A). In contrast, zinc oxide enhanced levels of Zn, Fe, Cu, Mn, and Mg in Premium (Table 2, Figure 2). Generally, Se/Zn treatments did not significantly impact growth parameters and concentrations of soluble solids and protein (Table 2, Figure 1).

In this investigation, we have focused on pea, as the crop represents an important staple legume and has been emphasized as a major player to assure food security worldwide, especially due to its high protein concentration. Peas have also been found to be able to accumulate Se and Zn more efficiently compared to cereals, which is likely also related to their high protein concentration [42,43]. The pea varieties included in the study were chosen due to their reported high yield. Foliar application has been described as an efficient method for biofortification, bypassing the effect that soil may have on the bioavailability of examined minerals [34]. While Zn is an indispensable mineral vital for plants [44], the essentiality of Se to higher plants is yet to be established. However, at low concentrations, Se exerts positive impact on plant development and yield, and improves abiotic stress resilience [45]. Regarding the forms of application, selenate was reported to exhibit a high efficiency for foliar uptake [46], while zinc oxide was chosen as recommended by the agro-industry. Our recent findings showed a positive effect of foliar-applied selenate and zinc oxide on the aggregation of phenolics in pea seeds [36]. In addition, upon these applications, total condensed tannins did not correlate with a lower accretion of selected minerals, macronutrients and plant bioactive constituents in pea seeds, which is encouraging in light of agronomic biofortification [34], as tannins have been reported to potentially hamper the nutritional value of many food items, including legumes, by interfering with the digestion and bioavailability of nutrients, including especially divalent minerals [47].

Selenate and zinc oxide did not specifically influence growth parameters (Table 2, Figure 1A–E), which is in line with our pot study (other than in part for the number of seeds/pod upon Zn treatment) [35]. Similar findings were also found by earlier foliar biofortification of peas with selenate and zinc sulphate [43], as well as with selenate and selenite [48]. However, Pandey et al. [49] reported a positive impact of foliar-applied zinc sulphate on the yield parameters of field pea.

Selenate and zinc oxide did not produce beneficial changes on the levels of soluble solids in pea seeds (Table 2, Figure 1F), which is consistent with our pot study [34]. However, previous studies did report increased concentrations of total sugars in seeds of pea [49], black gram [50], and common bean [51] following zinc sulphate application. Soluble solids are mostly simple carbohydrates that are needed for seed development [49]. Carbohydrate metabolism may be positively or negatively affected by selenium, depending on its employed concentration and form, as well as the plant developmental stage, as emphasized previously [52]. The negative impact of Se applications on soluble solids in pea seeds may be related to the lowered content of chlorophyll in seeds [34]. Regarding Zn, its lower status was linked to reduced photosynthesis and activity of sucrose synthase, which consequently impaired sugar synthesis in seeds, as emphasized earlier [49].

Selenate and zinc oxide did not improve protein levels in pea seeds (Table 2, Figure 1G). In contrast, our pot study showed that these applications enhanced protein concentration in Premium seeds [34]. Other research findings have revealed both beneficial and non-significant influences of foliar-applied selenate and/or zinc sulphate on protein levels in pea seeds [42,43,48,53]. The present study further found that Se and Zn concentrations were not significantly correlated with protein concentrations in Ambassador and Premium seeds (Figure 3). However, under pot experimental conditions, the concentration of Zn exhibited a positive and significant correlation with protein levels in both seed varieties [34]. Zinc may influence protein synthesis according to its involvement in DNA/RNA metabolism, chromatin condensation, and gene expression [54]. As previously reported, with respect to Se, its major species in pea seeds following foliar application of selenate was selenomethionine, with relevant antioxidant activity and possible benefits to human and animal health [35,55].

The detected concentration of total Se in the experimental soil (Table 1) is considered deficient according to Gupta and Gupta [56]. Ambassador variety in general showed higher concentrations of Se in seeds than Premium variety upon selenate applications (Figure 2A). In contrast, our pot study demonstrated greater accumulation of Se in the seeds of Premium rather than Ambassador variety following these applications [35]. The positive impact of foliar-applied selenate on the level of Se in pea seeds was also reported by other research [42,43,48,55]. The positive linear relationship between selenate application dose and seed Se level (Appendix A) is in accordance with earlier research on pea [35,42,48].

The concentration of Zn determined in the experimental soil (Table 1) is considered to be low, according to previous reports [57,58]. Zinc oxide applications improved Zn concentration only in Premium seeds (Figure 2B). However, the correlation between Zn application dose and Zn concentration in Premium seeds was not significant (Appendix A). In contrast, our pot study demonstrated that zinc oxide applications had no positive influence on Zn concentrations in Ambassador and Premium seeds [35]. However, Pandey et al. [49] revealed that foliar application of zinc sulphate enhanced Zn concentration in pea seeds. These findings suggest that zinc sulphate should be used in further investigations on foliar Zn biofortification of peas. However, other and novel Zn forms may also deserve further research, such as ZnEDTA and a Zn–glycine complex, which demonstrated lower phytotoxicity than zinc sulphate [59], allowing applications at a wider dose range.

Selenate, unlike zinc oxide, improved, although in part and not consistently, the concentrations of Fe, Cu, and Mn, while both Se and Zn applications had no favourable effect on Mo levels in pea seeds (Table 2, Figure 2C–F). In contrast, our pot study showed that these applications did not enhance concentrations of Fe, Cu, and Mn in pea seeds [36]. Likewise, other research has demonstrated that foliar-applied selenate and zinc sulphate did not positively influence the level of Fe in the seeds of pea [42,43,53]. Similarly, Kayan et al. [60] reported a non-significant influence of foliar-applied zinc chelate/sulphate on Fe levels in chickpea seeds.

Selenate and zinc oxide failed to significantly impact macroelement concentrations except for that of Mg in Premium seeds upon Zn applications (Table 2, Figure 2G–J). In contrast, our pot study revealed that these applications did not significantly affect the concentrations of Ca, Mg, K, and Na in pea seeds [34]. Likewise, previous research demonstrated that foliar-applied selenate and zinc sulphate had no beneficial effects on levels of Ca and Mg in pea seeds [42,43,53].

The significant relationships observed between minerals in pea seeds (Figure 3) are consistent in part with our earlier findings reported under pot experimental conditions [34,35,36], namely the positive correlations between Zn and Fe, Cu, Mn, and Mg, as well as the negative one between Se and Zn. Several aspects regarding interactions between accumulation of trace elements were discussed by our previous work [36]. A number of factors may impact the association between minerals in plants, including plant species/genotype, fertilizer dose, form and way of application, conditions of cultivation, soil properties, as well as antagonism and/or synergism between various elements, as emphasized by Malka et al. [34].

Finally, consuming 33 g/day of pea biofortified with Se at 50 g/ha and 266 g/day of pea bio-fortified with 375 g of Zn/ha could provide 100% of the RDA (55 μg) for Se and RDA (9.5 mg) for Zn in adults, respectively (Table 3).

## 5. Conclusions

Taken together, the present investigation demonstrated that selenate and zinc oxide had no distinct adverse impact on the growth parameters of pea, in accordance with an absence of toxic effects. Zinc oxide improved seed Zn accumulation only in one pea variety, and as similar effects were encountered in our pot experiment, more bioavailable alternative forms of Zn, namely zinc sulphate, should be employed in future investigations of peas. Contrarily, selenate considerably increased seed Se accretion in both varieties in a dose-dependent manner. Small portions of pea biofortified with 50 g of Se/ha and 375 g of Zn/ha would cover daily intake recommendations for Se and Zn, respectively; however, lower selenate doses could be envisioned in further research. In addition, the use of nanoparticles for foliar applications has been receiving some attention, but no results on peas are currently available. Foliar Se biofortification of pea could be an encouraging approach to improve human micronutrient supply.

## Figures and Tables

**Figure 1 foods-12-01286-f001:**
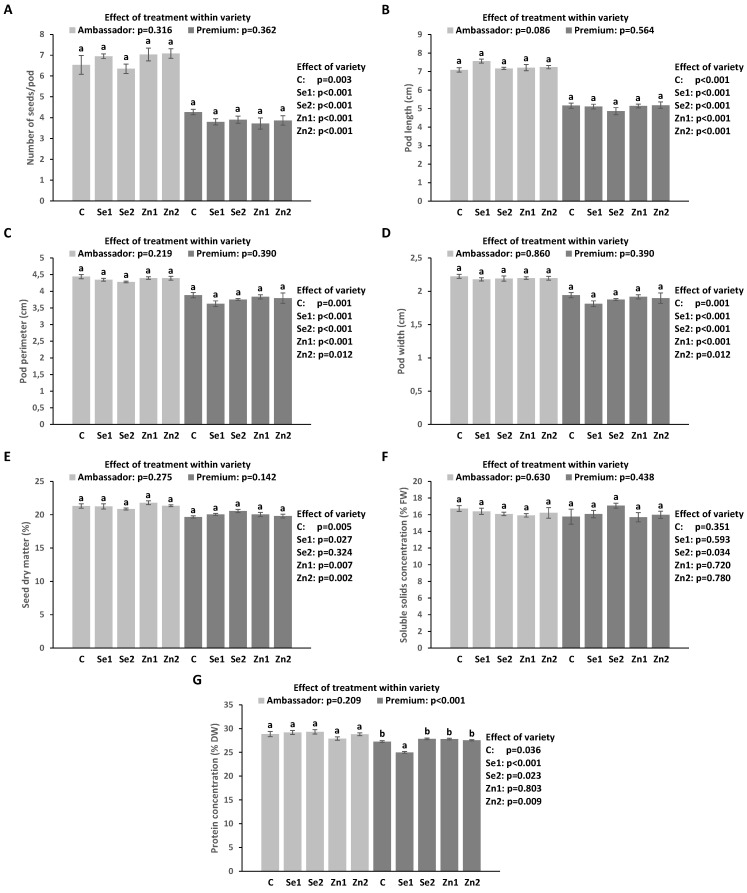
Impact of foliar Se and Zn treatments and variety on the number of seeds/pod (**A**), pod length (**B**), pod perimeter (**C**), pod width (**D**), seed dry matter (**E**), and concentrations of soluble solids (**F**) and protein (**G**) in seeds. C (control): no applied Se and Zn; Se1: 50 g; Se2: 100 g; Zn1: 375 g; Zn2: 750 g; all per ha; mean ± SD; n = 4 replicates. Bars without identical lower letters differed significantly within variety. *p*-values given on the right part of the graph indicate the impact of treatment across varieties (Ambassador compared to Premium).

**Figure 2 foods-12-01286-f002:**
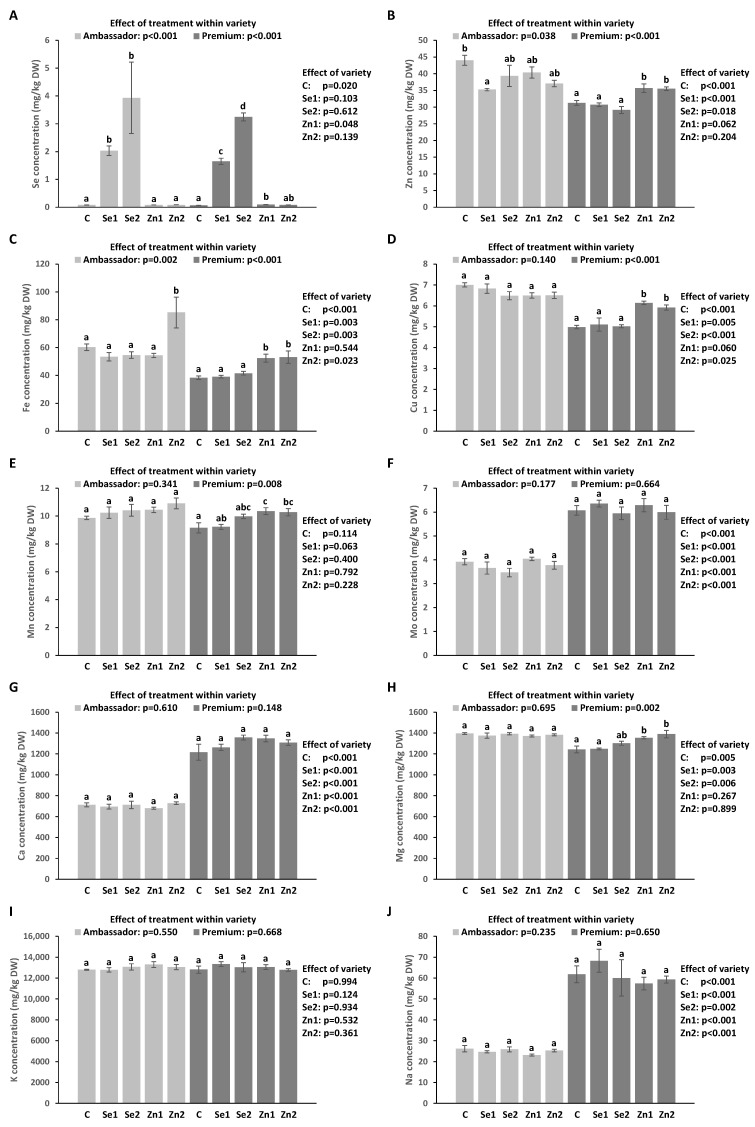
Foliar-applied Se and Zn and variety affect the concentration of Se (**A**), Zn (**B**), Fe (**C**), Cu (**D**), Mn (**E**), Mo (**F**), Ca (**G**), Mg (**H**), K (**I**), and Na (**J**) in seeds. C (control): without Se and Zn; Se1: 50 g; Se2: 100 g; Zn1: 375 g; Zn2: 750 g; all per ha; mean ± SD; n = 4. Bars that do not share an identical lower letter differ significantly within variety. *p*-values on the right part of the graph display the impact of treatment across varieties (Ambassador compared to Premium).

**Figure 3 foods-12-01286-f003:**
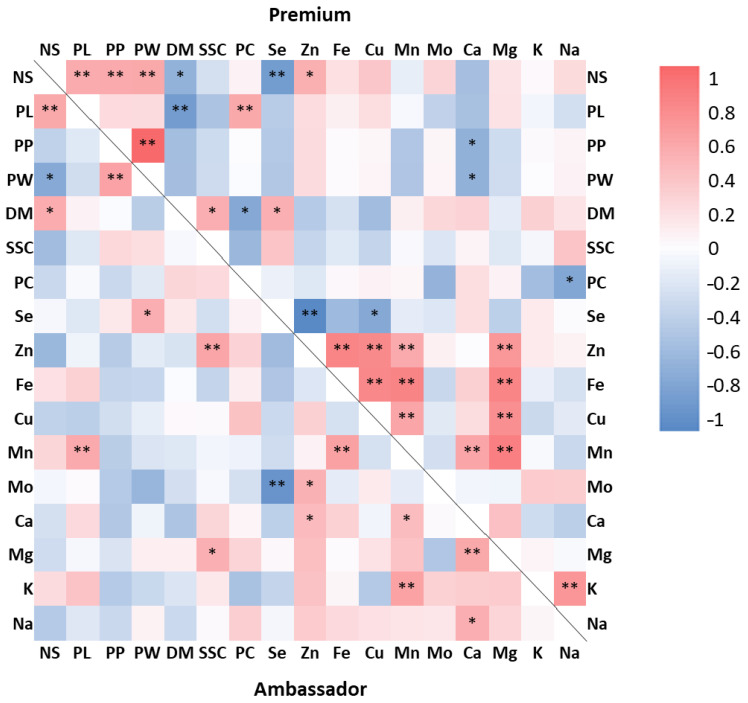
Correlation coefficients (Pearson) between growth parameters and mineral and macronutrient concentrations in seeds determined for the two pea varieties studied (Ambassador, Premium). NS: number of seeds/pod; PL: pod length; PP: pod perimeter; PW: pod width; DM: seed dry matter; SSC: soluble solid concentration; PC: protein concentration. Correlations below and above the black line indicate Ambassador and Premium, respectively. Significance level, * *p* < 0.05, ** *p* < 0.01.

**Table 1 foods-12-01286-t001:** Physical and chemical variables of tested soil, mean monthly temperature (air) and total monthly rainfall (April–June) reported in the experimental area in Nitra in 2014.

**pH (H_2_O)**	**pH (KCl)**	**N**	**P**	**K**	**Ca**	**Mg**	**Zn**	**Se**	**C_ox_ (%)**	**Humus (%)**	**Soil Type**
		**mg kg^−1^**									
7.55	6.36	13.3	253	285	5630	364	2.47	0.08	1.39	4.01	Gleyic fluvisol
**Average air temperature (°C) and total rainfall (mm) in brackets**											
April				May					June		
12.4 (48.9)				15.2 (57.6)					19.3 (52.5)		

C_ox_: oxidizable carbon.

**Table 2 foods-12-01286-t002:** Combined ANOVA for the effects of treatment and variety on number of seeds/pod, length, perimeter and width of the pod, seed dry matter, and levels of soluble solids, protein, Se, Zn, Fe, Cu, Mn, Mo, Ca, Mg, K, and Na in seeds.

	Treatment	Variety	Treatment × Variety
DF	4	1	4
Number of seeds/pod	NS	<0.001	NS
Pod length (cm)	NS	<0.001	NS
Pod perimeter (cm)	NS	<0.001	NS
Pod width (cm)	NS	<0.001	NS
Seed dry matter (%)	NS	<0.001	NS
Soluble solids (% FW)	NS	NS	NS
Protein (% DW)	0.002	<0.001	<0.001
Se (mg/kg DW)	<0.001	NS	NS
Zn (mg/kg DW)	0.002	<0.001	0.002
Fe (mg/kg DW)	<0.001	<0.001	0.011
Cu (mg/kg DW)	0.007	<0.001	<0.001
Mn (mg/kg DW)	0.004	0.004	NS
Mo (mg/kg DW)	NS	<0.001	NS
Ca (mg/kg DW)	NS	<0.001	NS
Mg (mg/kg DW)	0.004	<0.001	0.001
K (mg/kg DW)	NS	NS	NS
Na (mg/kg DW)	NS	<0.001	NS

DF: degrees of freedom; NS: not significant.

**Table 3 foods-12-01286-t003:** Selenium/Zn intake and fraction (%) of recommended dietary allowance for Se/Zn (% RDA) obtained from 100 g of pea seeds.

**Se**				
**Variety**	**Se** **Treatment** **(g of Se/ha)**	**Se Intake** **from 100 g (μg/day)**	**% RDA** **from 100 g (USDA *)**	**% RDA** **from 100 g (EFSA *)**
Ambassador				
	0	8	15	12
	50	203	369	290
	100	393	715	562
Premium				
	0	7	12	10
	50	165	300	235
	100	325	591	464
**Zn**				
**Variety**	**Zn** **Treatment** **(g of Zn/ha)**	**Zn Intake** **from 100 g (mg/day)**	**% RDA** **from 100 g (USDA *)**	**% RDA** **from 100 g (EFSA *)**
Ambassador				
	0	4.40	46	38
	375	4.04	42	35
	750	3.71	39	32
Premium				
	0	3.13	33	27
	375	3.57	38	31
	750	3.55	37	31

* 55 and 70 μg are the RDA and AI (adequate intake) set by the USDA and EFSA, respectively, for Se; * 9.5 and 11.5 mg are the RDA and AI, representing averages for males and females for Zn, according to USDA and EFSA, respectively.

## Data Availability

No new data were created or analyzed in this study. Data sharing is not applicable to this article.

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
