# Peer review of "Foliar Selenate and Zinc Oxide Separately Applied to Two Pea Varieties: Effects on Growth Parameters and Accumulation of Minerals and Macronutrients in Seeds under Field Conditions"

_foods, 2023, doi:10.3390/foods12061286_

Round 1
Reviewer 1 Report
Very clear experimental design and equally clearly written manuscript on the effects of foliar application of Zinc and Selenium to 2 high yielding, dark-seeded pea varieties grown on Gleyic fluvisol soil plots.
Please consider discussing the potential differences between zinc compounds as foliarly applied fertilizers (zinc sulfate for instance has demonstrated significant increases of grain zinc content), and relate to the composition of the zinc product used in your field trial (oxide % and chelate %). Were these nano particles (what is the particle size of the ZnO in the commercial product used? what is known about the chelated zinc form used in this compound? how can this be optimized for crops such as peas?
Please use a different reference for the stated global prevalence of Selenium deficiency (15%). I could not find this figure (or the 1 in 7 people estimate) in the reference you used. Other edits in annotated pdf file (attached).

Reviewer 2 Report
The review of “Foliar Selenate and Zinc Oxide separately applied to Two Pea Varieties: Effects on Growth Parameters and Accumulation of Minerals and Macronutrients in Seeds under Field Conditions” for Foods MDPI. The topic of manuscript fits within the scope of the journal and results can be considered of interest in order to improve human micronutrient-supply trough the biofortification of legumes with selenium and zinc.
The experiments appear to have been correctly designed and performed, the methods used are appropriate, and the manuscript presents a considerable amount of experimental results, which are in general coherent, and clearly presented, described and discussed. Furthermore, the conclusions of the work are sound and based on the results presented in the manuscript. Therefore, in my opinion, no serious criticisms can be raised on this study.
I have just a few remarks, which I give under author’s consideration:
1. L97-98: Please rephrase the sentence.
2. L100: Please check the form of selenium. I assume that “selenite” is wrong.
3. Figures 1: The sentence “Bars not having and identical number differed significantly within variety” should be rephrased.
4. L299: The phrase “all treatments but Se2” is confused.
5. Table 3: Does “Se intake from 100 g” presented on fresh or dry matter? As a rule, peas are not consumed in a dry form, so it is more expedient to make calculations in the table using data on the content of selenium in terms of fresh weight.
